# Microstructure and Microhardness Evolution of Additively Manufactured Cellular Inconel 718 after Heat Treatment with Different Aging Times

**Juan Manuel Salgado-Lopez** [1] , **Enrique Martinez-Franco** [1] , **Celso Cruz-Gonzalez** [1] , **Jorge Corona-Castuera** [2] and **Jhon Alexander Villada-Villalobos** [3,*]

1   Center for Engineering and Industrial Development (CIDESI), Av. Playa Pie de la Cuesta No. 702, Desarrollo San Pablo, Querétaro 76125, Mexico
2   CIATEQ A.C., Querétaro 76246, Mexico
3   National Council for Science and Technology, Center for Engineering and Industrial Development (CONACYT–CIDESI), Carretera Estatal 200, km 23, Querétaro 76265, Mexico
*   Correspondence: jhon.villada@cidesi.edu.mx

**Abstract:** The manufacture of cellular structures using high-performance materials is possible thanks to the additive manufacturing of metals. However, it is well known that the mechanical and microstructural properties of metals manufactured by this technique do not correspond to those of the same metals manufactured by conventional methods. It is well known that the mechanical properties depend on the direction of manufacture, the size of the pieces, and the type of cell structure used. In addition, the effect of heat treatments on parts manufactured by additive manufacturing differs from parts manufactured by conventional methods. In this work, the microstructure and microhardness of cellular structures of Inconel 718, manufactured by additive manufacturing under heat treatments with different aging times, were evaluated. It was found that the time of the first aging impacts the microhardness and its homogeneity, affecting the microstructure. The highest hardness was obtained for an aging time of 8 h, while the lowest standard deviation was obtained at 10 h. Finally, it is shown that the aging time influences a more homogeneous distribution of the elements and phases.

**Keywords:** additive manufacturing; Inconel 718; heat treatment; cellular structures; metamaterials





## 1. Introduction

Nickel-based superalloys are employed in critical components under severe temperature and corrosion conditions. Among these alloys, Inconel 718 (IN718) is one of the most widely used in the aerospace, aviation, and oil industries [1–5]. Therefore, it is selected for continuous research to improve its performance or to find a new manufacturing process of Inconel with the same or better properties as the original alloy. On the other hand, additive manufacturing (AM) is one of the methods to which the industry has paid a lot of attention due to its capabilities and the many possibilities to fabricate complex shapes. In this sense, this technology is very appropriate for manufacturing cellular materials, one of the most intricate geometries, as well as promising for the lightweight, efficient, and superior future design of engineering structures. Hence, the additive manufacturing of complex components of IN718 is a current topic of research [6].

However, some issues must be addressed before AM reaches its full potential: poor ductility, surface roughness, distortion, anisotropy, pores, and high residual stresses [6]. Many of those problems are related to temperature gradients from the complex heating and cooling cycles of layer-by-layer manufacturing, which usually introduces many undesired microstructure features in the as-built alloys, such as composition inhomogeneity, segregation, and texture [1–6].

The most reported results in the technical literature are related to the correlation of the process parameters with the reduction of such problems for improving the mechanical

properties, but that is a partial solution [7]. Therefore, post-processing treatments are required. Furthermore, there are some other unsolved issues in the case of the microstructure of AM, such as pores, lack of fusion, variation in the grain size, texture, and segregation of some elements or phases at the molten pool boundaries [7–12].

One of the main problems related to the IN718 microstructure is the precipitation of detrimental phases, such as secondary metallic carbides ($M_{23}C6$), $\delta$-$Ni_3Nb$, and Laves phases. In the case of the microstructure of IN718 fabricated by AM, the literature reported $\delta$ and Laves phases in the molten pool boundaries driven by the segregation of some elements. It has been identified as the reason for the low ductility and lower toughness of additively manufactured IN718 [7,13–15]. According to the literature review reported by Wang et al. [1], the $\delta$ phase also precipitates at grain boundaries in a needle-like form. Laves phases are irregularly shaped phases formed due to Nb segregation with the other alloying elements with a typical composition of $(Ni, Fe, Cr)_2(Mo, Nb, Ti)$. Regarding the grain structure, selective-laser-manufacturing-produced IN718 parts generally present a fine columnar dendrite structure with internal micro-segregation in interdendritic regions [1].

On the other hand, the detrimental phases can be dissolved in the matrix by proper heat treatments. According to the literature [8,13–15], the heat treatment for IN718 promotes the nucleation and growth of the reinforcement phases, such as $\gamma'$ and $\gamma''$. Moreover, it controls the size and concentration of secondary phases, such as $\delta$ and Laves phases, and carbides. For these purposes, some standards [16,17] propose a heat treatment consisting of three steps: (1) solid solution treatment (ST) (900–1100 °C for 60 min) to establish a single-phase FCC $\gamma$ matrix and dissolve the constituents and precipitates; (2) first aging treatment (700–760 °C for 480 min), to precipitate the primary $\gamma'$ and $\gamma''$ phases; (3) second aging treatment, (620–650 °C for 480 min), to promote the formation of fine $\gamma'$ and $\gamma''$ precipitates and the formation of primary carbides in grain boundaries. However, it is well known that the effect of this conventional heat treatment is different when it is applied to AM parts [17,18].

Cellular materials and metamaterials are important for the industry because they combine being lightweight with good mechanical properties. Furthermore, they offer the possibility of improving porosity in components where the unit area is essential for improving performance [18,19]. For instance, in the construction, aerospace, and chemical industries, AM is the best technology for manufacturing cellular materials or metamaterials. However, in the case of cellular IN718, there is too little information in the technical literature. For instance, Hu et al. reported on additively manufactured pentamode meta IN718, proposed for the thermal stress accommodation of alkali metal heat pipes. They reported that metamaterials accommodated thermal stresses in sodium/IN718 heat pipes better than tubes without meta IN718. Moreover, they informed that, with pentamode metamaterial reinforcement, thermal stresses in the heat pipes were reduced [20]. Nevertheless, no information was given about either the heat treatment or the microstructure evolution.

Huynh et al. fabricated hybrid cellular structures and performed heat treatment by solution annealing followed by aging/precipitation hardening in accordance with AMS 5662 to achieve the desired morphology and distribution of the $\gamma''$ precipitate structure in the IN718 material [21]. The microstructure characterization reported that the $\gamma''$ precipitates are the most relevant. No further studies were performed.

It must be kept in mind that the reported microstructure of the additive manufacturing of "bulk" IN718 "as-built" consists of dendritic and columnar grain structures. Moreover, the segregation of niobium and molybdenum at the melt pool boundaries (MPBs) is well known; these elements are essential for hardening. However, when they are at the grain boundaries, they impair the mechanical properties of IN718. Therefore, heat treatments for "bulk" AM IN718 are investigated to modify the microstructure, especially the segregation of the elements, to obtain mechanical and corrosion properties similar to the same alloys manufactured by conventional methods.

Teng et al. [22] reported the microstructure evolution and tensile properties of bulk IN718 in the as-built condition and after various heat treatments. They showed the typical

melt tracks and melt pool boundaries. At the same time, their results indicate that the as-built microstructure consisted of dendritic and cellular structures, while small $\gamma''$ (observed by transmission electron microscopy) phases and acicular $\delta$ phases formed after heat treatment.

The cellular structures of other metals have also been studied. For example, Yang and co-workers reported the fabrication of cellular samples made from Ti6Al4V using the "electron beam melting" technique. The surface appearance after manufacturing is only reported as a microstructure evaluation, and it is the thermal dissipation during the fabrication process that affects the geometrical accuracy of the structures and creates crack initiation sites [23].

Despite the research done on bulk IN718, there is little information about the cellular IN718 microstructure and the effect of heat treatment on this material. Therefore, this work aims to study the microstructure and microhardness of additively manufactured cellular IN718 and the evolution of the microstructure with different heat treatments.

Cellular samples were heat treated in a vacuum furnace, including a solution treatment and two aging steps. The time for the first aging step varied from 6, 8, and 10 h to study the effect of this time on the microstructure and microhardness. The solution treatment and second aging step were the same for all samples. It was found that the longer the first aging step, the greater the homogeneity in the microstructure and microhardness. The highest microhardness corresponds to 8 h.

## 2. Materials and Methods

Cellular architectures of 50 mm × 50 mm × 50 mm (Figure 1) were built using the laser powder bed fusion (L-PBF) technique in an EOSINT M 280 (EOS GmbHElectro Optical Systems, Krailling/Munich, Germany). The unit cell used to reproduce the cellular architecture is shown in the inset of Figure 1. The diameter of the wires (0.584 mm) was selected to obtain a relative density of 10% compared to a solid part of the same volume. The parameters for the central area were: laser power of 285 W, scanning speed of 960 mm/s, and hatch distance of 0.088 mm. For the lower area, they were: laser power of 76 W, scanning speed of 785 mm/s, and hatch distance of 0.08 mm.

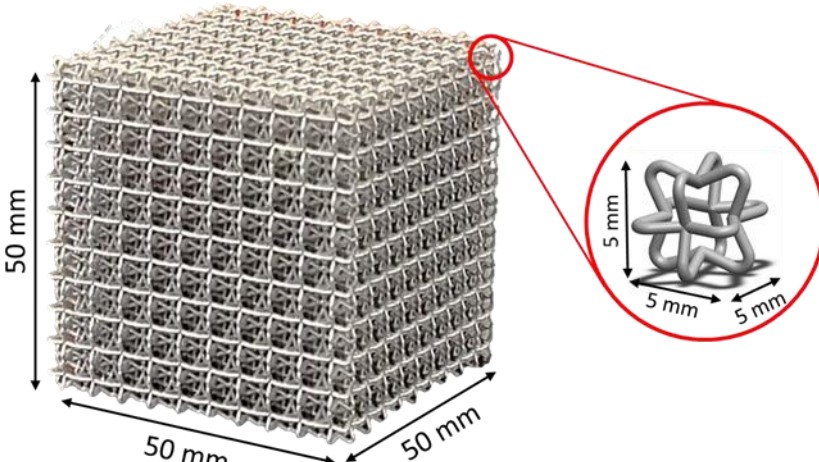

**Figure 1.** Cellular architecture of the samples.

IN718 pre-alloyed powders produced by gas atomization (EOS GmbH, Krailling/Munich, Germany) were used as feedstock. The chemical composition of the IN718 powder is shown in Table 1. It was studied using atomic absorption, Leco CS 200 equipment, and wavelength-dispersive X-ray spectroscopy (WDS), Jeol JXA8530F.

Scanning electron microscopy was used to study the size distribution of the IN718 powder. Results are shown in Figure 2. The Gaussian distribution, where d10, d50, and d90, was 19.23 μm, 31.15 μm, and 55.89 μm, respectively. The average size is 38.15 μm.

**Table 1.** IN718 powder composition obtained from different analysis methods.

| Element | EOS Report | Atomic Absorption | WDS |
|---|---|---|---|
| Carbon | 0.08 % Max | 0.04% | – |
| Sulfur | 0.015% Max | 0.002% | – |
| Chromium | 17.0–21.0% | 18.30% | 17.18% |
| Molybdenum | 2.8–3.3% | 2.60% | 2.60% |
| Titanium | 0.65–1.15% | 0.89% | 1.07% |
| Aluminum | 0.20–0.80% | 0.36% | 0.75% |
| Cobalt | 1.0% Max | 0.04% | – |
| Copper | 0.3% Max | 0.01% | – |
| Silicon | 0.35% Max | <0.000 | – |
| Manganese | 0.35% Max | 0.02% | – |
| Nickel | 50–55% | 51% | 52.41% |
| Niobium | 4.75–5.5% | – | 6.81% |
| Iron | Remainder | 18% | 19.17% |

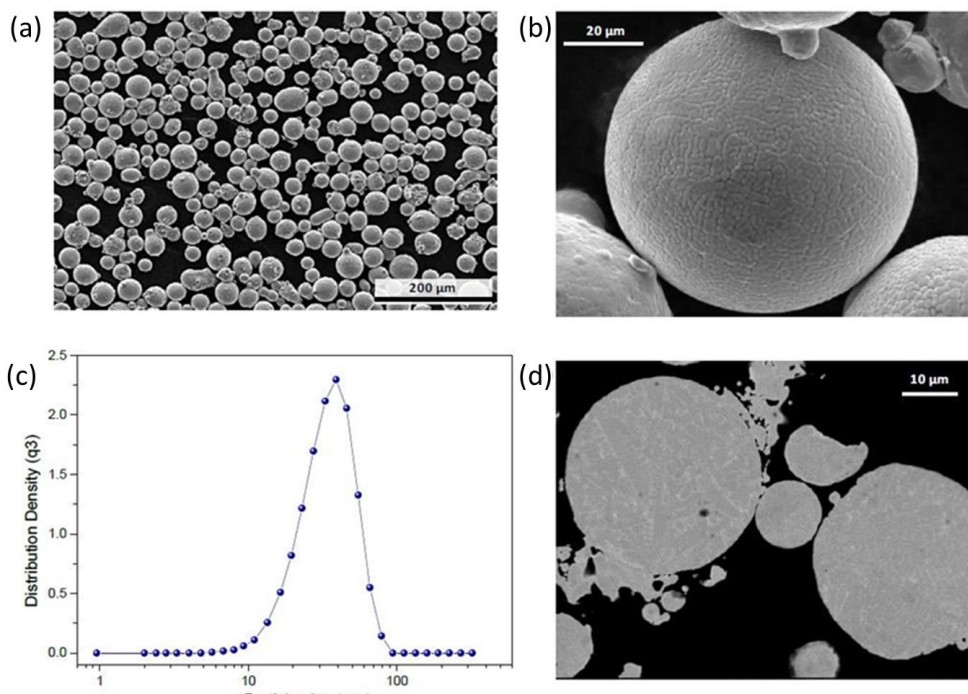

**Figure 2.** (**a**) IN718 EOS powder circularity, (**b**) Roughness of the powder particle, (**c**) Particle size distribution, (**d**) Backscattered image from the cross-section of the IN718 powder.

Heat treatments: It has been reported that the first aging treatment has the most relevant effect on the mechanical properties of the material, such as the ultimate tensile strength, yield strength, and hardness [8,13,15]. For this reason, in this work, three different heat treatments were applied to the AM-IN718, varying times for the first aging stage (6, 8, and 10 h), according to Table 2. The solution treatment and the second aging stage were the same for all the samples. The heating rate was 10 °C/min. The cooling rate after solution treatment was approximately −50 °C. After the first aging stage, the cooling rate was 0.83 °C/min, as stated in aerospace material specifications (AMS) standards [15,16]. The cooling rate after the second aging stage was 10 °C/min.

The samples were prepared for metallographic analysis following ASTM E3-11-2017. The cross-section cuttings were made along with the height (build direction) of the samples. After being mounted in conductive Bakelite, the test specimens were ground with sandpaper of a grade from 320 to 1200, sequentially, and polished with 1 μm alumina powder

slurry. Finally, the test samples were etched using Kalling's 2 as a reagent by 10 s immersion and rinsing with ethanol and deionized water to reveal the microstructure.

**Table 2.** Heat treatments applied to cellular structures of additively manufactured IN718.

| Heat Treatment | Solution Treatment | First Aging | Second Aging |
|---|---|---|---|
| T1 | 1050 °C/1 h | 720 °C/6 h | 620 °C/8 h |
| T2 | 1050 °C/1 h | 720 °C/8 h | 620 °C/8 h |
| T3 | 1050 °C/1 h | 720 °C/10 h | 620 °C/8 h |

The observation was made with a NIKON EPIPHOT 200 optical microscope (Tokyo, Japan). Phase identification was based on the morphology, location, and chemical composition of the precipitates observed by scanning electron microscopy (SEM) and energy dispersive spectroscopy (EDS). This methodology has been used before [17]. The SEM and EDS studies were carried out using an Oxford instrument model Ultimax 100 (Abingdon, UK) attached to an FE-SEM model JSMH-7200F (Tokyo, Japan). In this case, the microhardness of each condition was measured in the core branches of the samples using a TUKON 1202 Vickers microhardness tester with a 100 g load during 10 s of dwell time.

## 3. Results

### 3.1. Microstructure Evolution

The microstructures of the samples were observed using optical microscopy. Figure 3 shows the microstructure in the "as-built" condition. Meanwhile, the microstructures after heat treatment with 6, 8, and 10 h of aging are shown in Figures 4–6, respectively.

Figure 3 shows the typical AM solidified "pools" microstructure in cellular IN718. Columnar and cellular structures are rich in nickel, with interdendritic precipitates rich in niobium and molybdenum within MPBs with arc-shaped features, as is shown in the chemical analysis performed in such areas and is presented at the end of this section.

Although it is not shown, defects such as porosity, lack of fusion, and inclusions were found. Solidification pools have a similar appearance to those obtained from "bulk" samples reported in our previous research work [17] using the same process parameters. The main difference is the well formation of the melting pool tracks in the bulk samples, which is attributed to the more significant amount of powder in bulk compared to thin cellular architecture samples. Then, cooling is relatively faster in cellular than in bulk samples.

In this way, Figure 3b shows a SEM image of the microstructure of the material in the "as-built" condition. It is seen that the interdendritic concentration of the alloying elements is not homogeneous. It means that the size of the precipitates is different from one side of the melt pool boundary, respecting the other side.

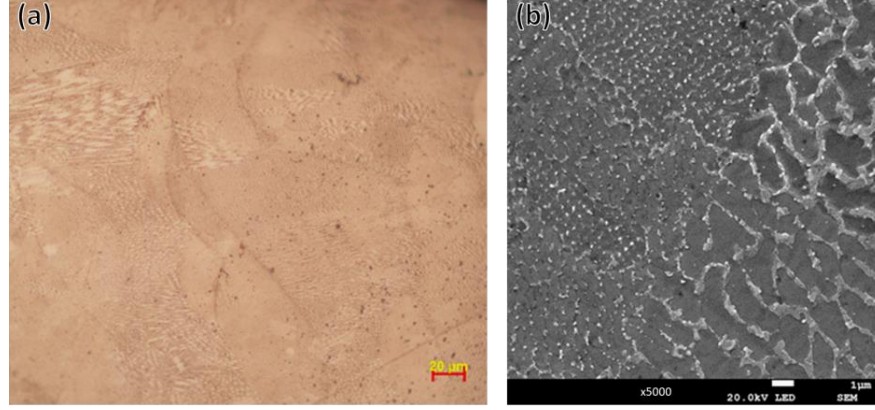

**Figure 3.** Microstructure of cellular IN718 in the "as-built" condition. (**a**) Optical microscopy at 500×. (**b**) SEM image at 5000×.

Figure 4 shows the microstructure of the cellular IN718 after 6 h of aging. It is seen that the "as-built" microstructure disappeared and is replaced by grains with different sizes and shapes surrounded by phases such as MC type metallic carbides, Laves, and δ phases. This behavior is clearly shown in Figure 4b, where it is seen that the phases at the grain boundaries are mainly Laves, δ, and primary carbides (MC, where M is Nb or Ti).

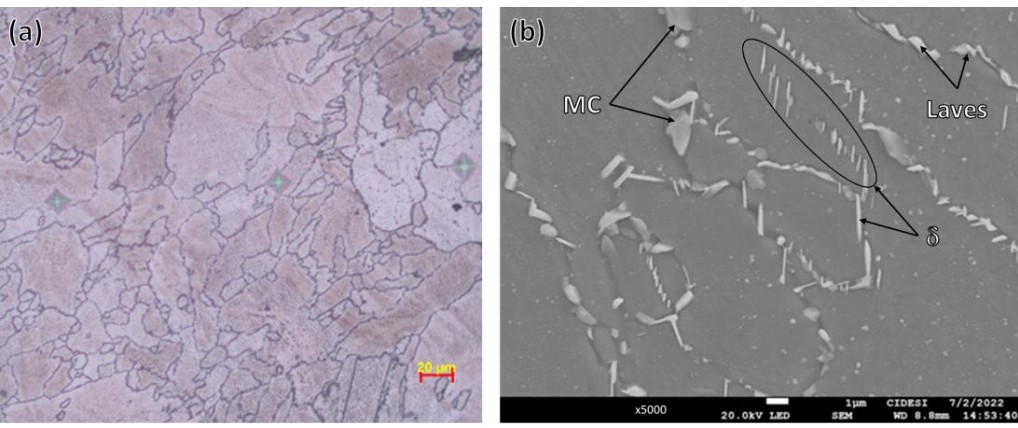

**Figure 4.** Microstructure of cellular IN718 after heat treatment with 6 h of aging. (**a**) Optical microscopy at 500×. (**b**) SEM image at 5000×.

Figure 5 shows the microstructure of cellular IN718 after 8 h of aging. There are grains with different shapes and sizes surrounded by Laves phases and metallic carbides. δ phases have almost disappeared. In addition, there are also subgrain structures within some grains. Moreover, there is evidence in the micrograph of grains with recrystallization twins and precipitates along the twin planes. In this way, Figure 5b shows a SEM image of the microstructure of the same condition, and it is shown that metallic carbides (MC) surround the grains, while subgrain structures are clearly defined.

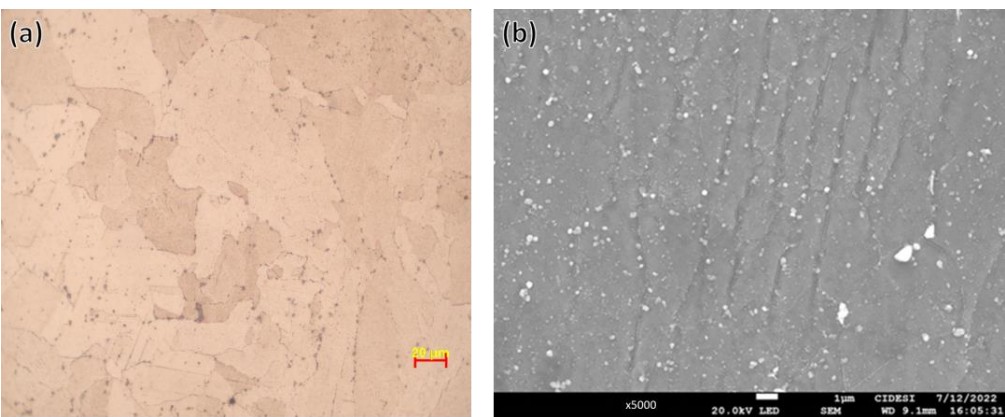

**Figure 5.** Microstructure at 500× of cellular IN718 after 8 h of aging. (**a**) Optical microscopy at 500×. (**b**) SEM image at 5000×.

Figure 6 shows the microstructure of cellular IN718 after 10 h of aging. It is seen that the grains remain with different shapes and sizes, as well as grain boundaries surrounded by metallic carbides (MC). Moreover, there are still subgrain structures. This is confirmed by Figure 6b, but in this image, it is clearly identified that the subgrain structure appears only in certain grains.

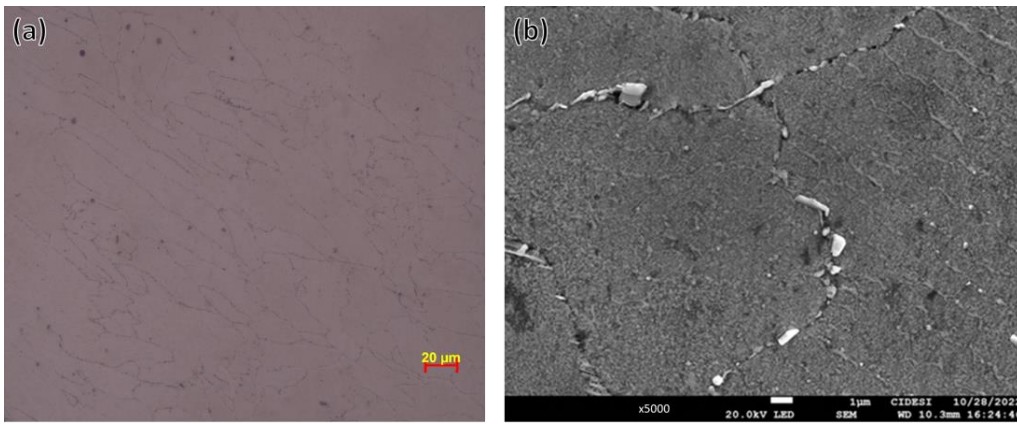

**Figure 6.** Microstructure of cellular IN718 after heat treatment with 10 h of aging. (**a**) Optical microscopy at 500×. (**b**) SEM image at 5000×.

In order to observe details of the subgrain structure, a higher magnification was made by SEM. Figure 7 shows the subgrain structure at 45,000× in the IN718 after 10 h of aging using backscattered electrons. In this Figure, the difference in contrast between the subgrains and the matrix of the grain is seen. According to the scale in this Figure, the subgrain's size is about 300 nm.

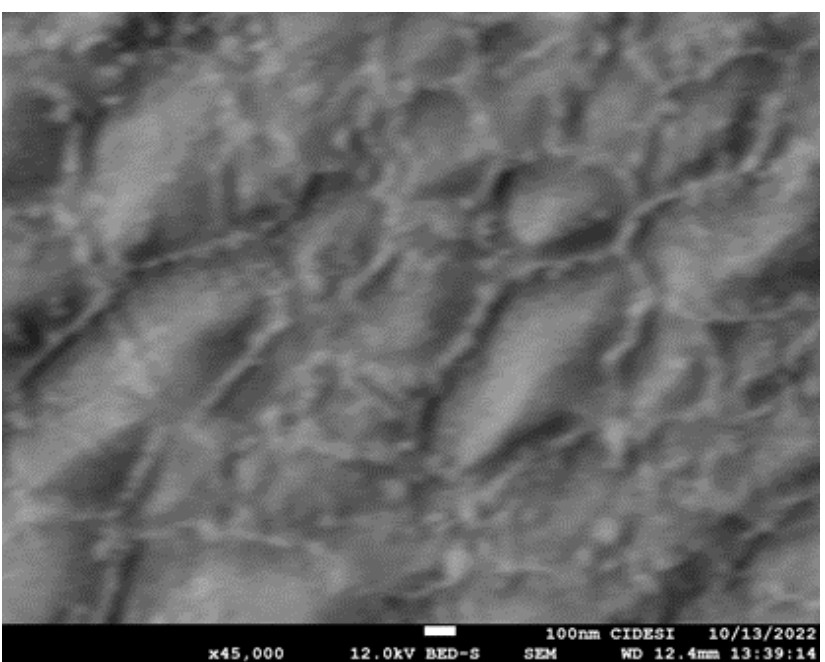

**Figure 7.** Microstructure at 45,000× of cellular IN718 after 10 h of aging using backscattered electrons. The subgrain structures can be observed.

Figure 8 shows the microstructure of cellular IN718 after 10 h of aging in the area where chemical mapping was carried out to distinguish the segregation of the alloying element of the material. Segregations at the grain boundary and the subgrain structures are determined with the chemical mapping results in Figure 9. At the grain boundary, there is a depletion of nickel and an increase in niobium, molybdenum, and titanium.

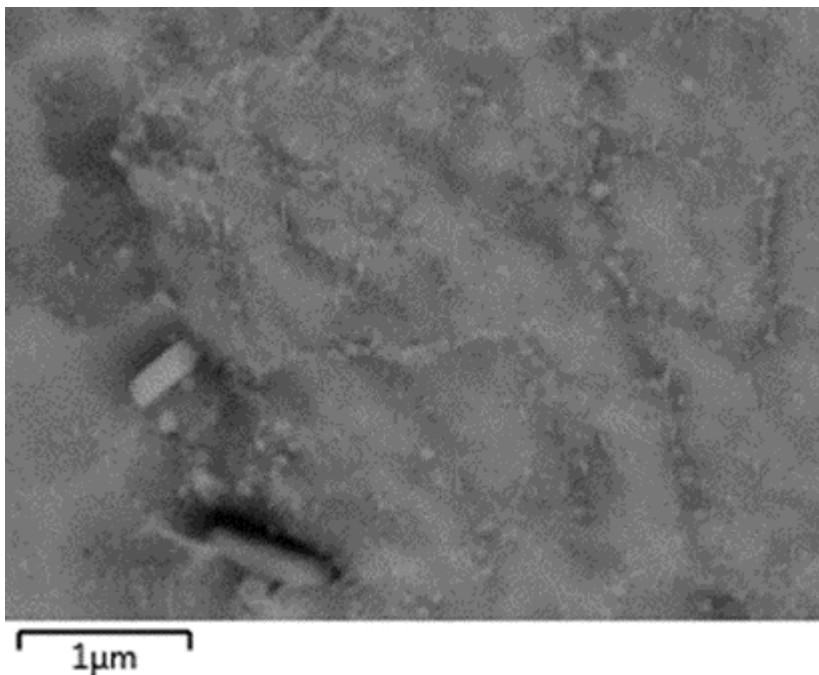

**Figure 8.** Microstructure at 7500× of cellular IN718 after 10 h of aging using backscattered electrons. This image was used for chemical mapping.

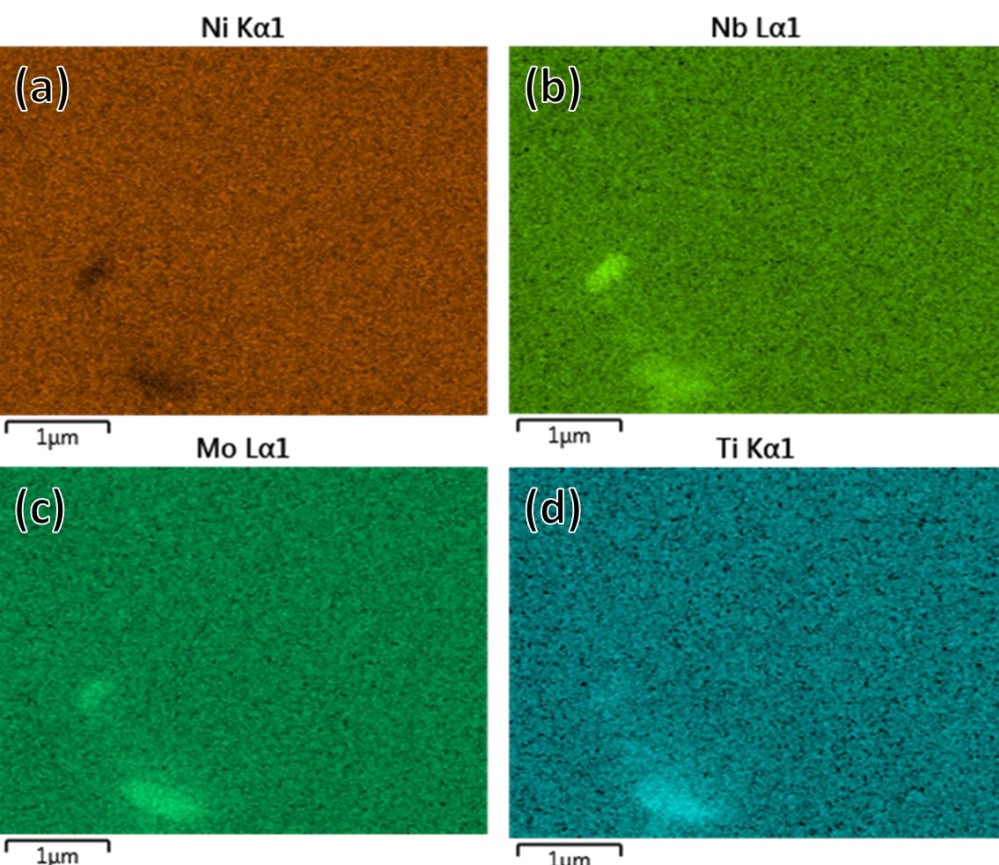

**Figure 9.** The images show the results of the chemical mapping. (**a**) Ni distribution, (**b**) Nb distribution, (**c**) Mo distribution, (**d**) Ti distribution.

Figure 10 shows the microstructure at 75,000× of cellular IN718 after 10 h of aging using backscattered electrons. The image was used for punctual EDS microanalysis in the

subgrain structure; the results are also shown in Figure 10. In the subgrain structure, there is a slight increase in molybdenum and niobium; meanwhile, there is a depletion of nickel.

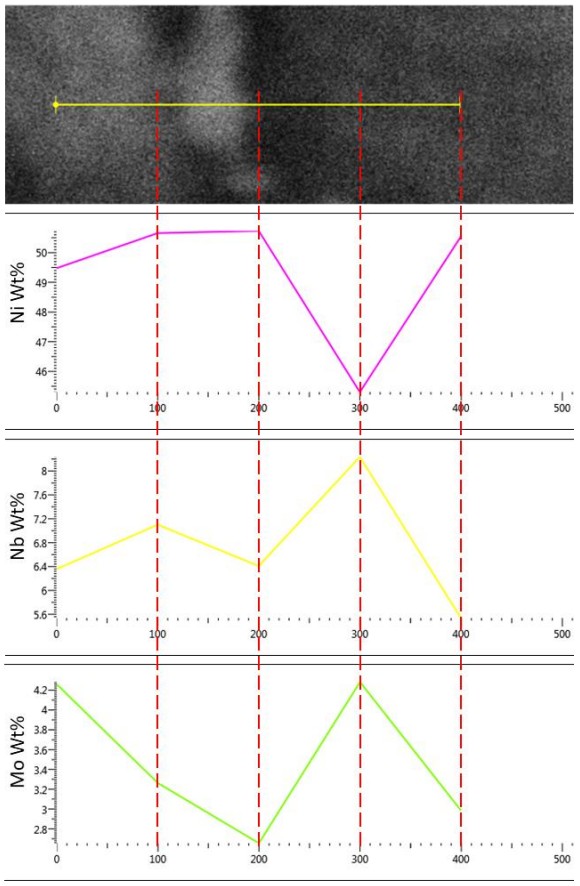

**Figure 10.** Microstructure of cellular IN718 after 10 h of aging and punctual EDS microanalysis in five positions along the marked line.

*3.2. Microhardness Evolution*

Table 3 shows the Vickers microhardness measurements in the samples of cellular IN718 prepared for metallography. The average and standard deviation were added to compare the results. Figure 11 shows the results of the comparison of the Vickers microhardness measurements carried out on the samples.

**Table 3.** Vickers microhardness measurements in HV0.1.

| Sample | Test 1 | Test 2 | Test 3 | Test 4 | Test 5 | Test 6 | Test 7 | Test 8 | Avg. | SD [1] | Variance |
|---|---|---|---|---|---|---|---|---|---|---|---|
| Reference | 353.5 | 318.2 | 368.2 | 368.2 | 368.2 | 368.2 | 368.2 | 368.2 | 368.2 | 368.2 | 368.2 |
| 6 h | 542.1 | 522.7 | 366.1 | 366.1 | 366.1 | 366.1 | 366.1 | 366.1 | 366.1 | 366.1 | 366.1 |
| 8 h | 518.6 | 551.9 | 368.3 | 368.3 | 368.3 | 368.3 | 368.3 | 368.3 | 368.3 | 368.3 | 368.3 |
| 10 h | 520.5 | 510.5 | 347.5 | 347.5 | 347.5 | 347.5 | 347.5 | 347.5 | 347.5 | 347.5 | 347.5 |

[1] Standard deviation.

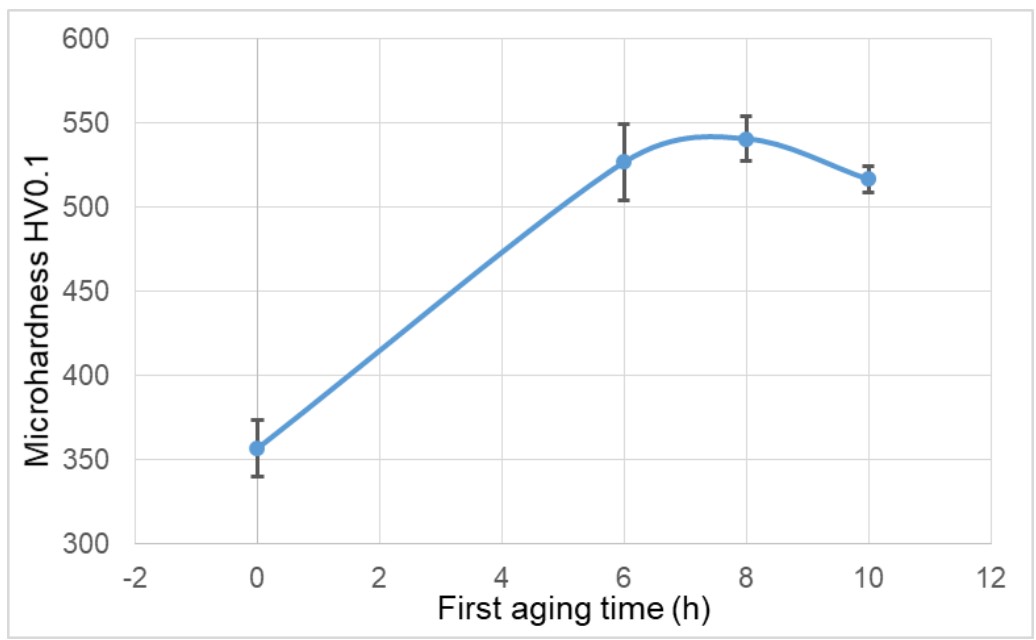

**Figure 11.** Results of the Vickers hardness measurements.

To better understand the effect of the first aging time on the microhardness, an ANOVA test of the data in Table 3 was performed. The results are shown in Table 4.

**Table 4.** ANOVA test of the data in Table 3.

| Source of Variation | SS | df | MS | F | *p*-Value | F Crit. |
|---|---|---|---|---|---|---|
| Between Groups | 178,398.57 | 3 | 59,466.19 | 197.26 | 0.00 | 2.95 |
| Within Groups | 8440.97 | 28 | 301.4630 | | | |
| Total | 186,839.54 | 31 | | | | |

Table 5 shows the results of a Tukey rank test also applied to the data in Table 3. This test complements the ANOVA test and allows us to define whether there is a significant difference between times for the first aging. According to the critical values of the Tukey table and the mean squared error of the data in Table 3, the honestly significant difference (HSD) value was calculated as 23.57. The values in Table 4 that are higher than the HSD represent a significant difference.

**Table 5.** Tukey test of the data in Table 3.

| | 0 h | 6 h | 8 h | 10 h |
|---|---|---|---|---|
| 0 h | | 170.1 | 183.9 | 159.9 |
| 6 h | | | 13.9 | 10.2 |
| 8 h | | | | 24.1 |
| 10 h | | | | |

## 4. Discussion

This work was carried out to identify the evolution of the microstructure and Vickers microhardness in cellular IN718 additively manufactured after 6, 8, and 10 h of aging. In this regard, Figure 3 shows the microstructure of cellular IN718 in the "as-built" condition. There are columnar and cellular structures with Nb-rich interdendritic areas within melt pool boundaries. It matches the microstructure reported in the literature for bulk AM IN718 [1–3,13–15].

According to the literature, the precipitates in the microstructure of the "as-built" sample with a high concentration of Nb are also identified as Laves-phase [1–3,13–15]. In this case, it is distinguished that the interdendritic areas are not distributed homogeneously in all grains; this is clear at high magnifications (Figure 3b). This behavior is evidence of the segregation of the alloying elements in some grains. Therefore, the sizes of the precipitates are different from one side of the melt pool boundary, respecting the other side (Figure 3b). This fact suggests a connection between the segregation in the "as-built" condition and the fact that the subgrain structures are not seen in all grains after heat treatment.

Figure 4 shows the microstructure after 6 h of aging; it is evident that the "as-built" microstructure changed to a different microstructure after heat treatment. δ phases and metallic carbides surround the grain boundaries, and subgrain structures start to appear in some grains. There was no evidence of melt pool boundaries or cellular or columnar structures. There is evidence of grains with recrystallized twins and small amounts of precipitates in the twinning grains. This evidence matches the difference in the amount of interdendritic segregation areas and suggests that the presence of the subgrain structures influences the recrystallization of the grains.

The technical literature has reported that the microstructure of additively manufactured IN718 in bulk samples is similar to that found in this work [1–3,13–15]. This fact is interesting because it has been stated that differences in solidification and cooling conditions lead to differences in the morphology of the microconstituents of the alloy. Nevertheless, in this case, the microstructure is similar to the one reported in much work. However, the solidification in a cellular structure is different than in a bulk structure [1,2,15,18,20–22,24–26].

In the case of 8 and 10 h of aging microstructures (Figures 5 and 6), there is a change in the size and shape of the phases surrounding the grain boundaries. It is clear that, as aging time increases, the subgrain structure becomes stable, and the precipitates at the grain boundaries change. The chemical mapping results show that niobium, titanium, and molybdenum carbides are found in the grain boundaries. Furthermore, the subgrain structures consisted of the segregation of niobium and molybdenum as was found by the punctual EDS microanalysis (Figure 10). This evidence matches the results reported in the literature [24–26].

In our previous investigation of bulk samples of IN718 obtained by AM [17], similar phase transformations were observed after performing the respective heat treatments. In bulk samples, heat transfer during the AM process allows a slower cooling rate than in relatively thin cell structure samples. Such transformations are slightly different in the amount observed.

The results of the microhardness measurements show that there is an increase in the hardness after aging. Interestingly, the average Vickers hardness in the "as-built" condition is slightly higher than that reported in the literature for AM bulk IN718. This conduct can be explained by the fact that the heating–cooling cycles in cellular Inconel are fewer than in bulk conditions [25,26].

The Vickers microhardness values after heat treatment are higher after 8 h of aging (540 HV0.1) than other conditions (526 for 6 h and 516 for 10 h of aging). These results suggest that, as the aging time increases, the average microhardness increases, but after 8 h, the microhardness decreases. This result indicates the effect of the $\gamma'$ and $\gamma''$ precipitation in the interior of the IN718 grains. This effect has been observed in other works, which concluded that this is mainly due to the recrystallization of the dislocation generated during AM [1,3,7,13,20–22,24–26]. In this work, evidence of recrystallization was observed in some grains of the microstructure after 8 h of aging (Figure 5). It is important to mention that a trend of the standard deviation of the Vickers microhardness measurements was the opposite. In other words, as aging time increases, the standard deviation decreases. This fact is crucial because it indicates a trend of homogenization of the alloying elements (especially Nb and Mo). This fact matches with the microstructure after 8 and 10 h of aging.

According to the analysis of variance (ANOVA) presented in Table 4, it can be seen that the value of $p$ (significance) is less than 0.05 since we have a confidence level of

95%. In addition, the value of the critical factor (2.95) is lower than the factor in the table (197.26). Therefore, it is possible to conclude that different aging times produce different microhardness. However, it is impossible to determine if this is true for all times analyzed. For this reason, the ANOVA analysis was complemented with a Tukey test.

According to the Tukey test reported in Table 5, it is possible to conclude that a variation of 6 to 8 h in the first aging time does not significantly affect the microhardness of the cellular IN718. However, a time of 10 h produces a significant decrease in microhardness.

**5. Conclusions**

This work was carried out to investigate the microstructure and microhardness evolution after the first aging heat treatment in additively manufactured cellular IN718. In addition, the results were compared to those reported in the literature for additively manufactured bulk IN718.

The microstructure of cellular IN718 in the "as-built" condition consists of columnar and cellular structures with Nb-rich interdendritic areas within melt pool boundaries, which matches with the microstructure reported in the literature for additively manufactured bulk IN718 [1–15]. Nevertheless, the average Vickers hardness in the "as-built" condition is slightly higher than that reported in the literature for AM bulk IN718.

Evidence of the segregation of the alloying elements in some grains suggests a connection between the segregation in the "as-built" condition and the fact that the subgrain structures are not seen in all grains after heat treatment. The evidence of grains with recrystallized twins and small amounts of precipitates in the twinning grains reinforces this theory.

As aging time increases, the subgrain structure becomes stable, and the precipitates at the grain boundaries change. As a result, the subgrain structures are not seen in all grains of the microstructure.

The Vickers microhardness values after heat treatment are much higher after 8 h of aging (540 HV0.1) than in other conditions (526 for 6 h of aging and 516 for 10 h of aging). This result indicates that microhardness increases as the aging time increases. Nevertheless, the standard deviation of the Vickers microhardness measurements showed a trend to reduce as aging time increases.

**Author Contributions:** J.M.S.-L.: conceptualization, formal analysis, writing—original draft preparation, methodology, and investigation; E.M.-F.: writing—review and editing, visualization, and investigation; C.C.-G.: supervision; J.C.-C.: funding acquisition and resources; J.A.V.-V.: funding acquisition, project administration, supervision, writing—review and editing, and formal analysis. All authors have read and agreed to the published version of the manuscript.

**Funding:** This research and the APC were funded by the National Council for Science and Technology of Mexico (CONACYT) in the year 2022, grant numbers 000000000319522 and 000000000285088.

**Data Availability Statement:** Not applicable.

**Acknowledgments:** J.A. Villada thanks CONACYT for the support through the "researchers for Mexico" program and the project 000000000319522. The authors also want to thank CIATEQ and the team of researchers involved in the CONACYT project 000000000285088 for providing the samples used in this work.

**Conflicts of Interest:** The authors declare no conflict of interest. The funders had no role in the design of the study; in the collection, analyses, or interpretation of data; in the writing of the manuscript; or in the decision to publish the results.

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
