# Peer review of "Microstructure and Microhardness Evolution of Additively Manufactured Cellular Inconel 718 after Heat Treatment with Different Aging Times"

_metals, doi:10.3390/met12122141_

Round 1
Reviewer 1 Report
The article submitted for review concerns the microstructure and microhardness of the Inconel 718 alloy produced in additive technology. This alloy has been heat treatment.
1. Abstract is written correctly, but I miss some details about the methodology. Please add 1-2 sentences about the conditions in which the samples were aged and what are the results of microhardness.
2. Specify the granularity of the powder, if you have it, attach a SEM photo of the powder.
3. What do classical metallographic procedures mean. Describe in detail the procedure for making a cross-section and the etching reagents.
4. Check the text carefully for typos, punctuation, and hardware data.
5. How the carbides were identified (were XRD studies carried out).
6. Please describe the microstructures in more detail, relate these descriptions to the manufacturing parameters.
7. Please relate your research results to other researchers in more detail.
8. There is no point in writing the size of the magnification (500x, 7500x etc.) It is enough that there is a scale bar in the photo. If someone prints a photo on half of a sheet of paper, the magnification data will not be correct. Therefore, remove this redundant data and leave only the scale bars.
9. How the linear EDS analysis was performed. The graphs look as if they were made from several points. You can't plot a graph from 5 points and represent it as linear EDS results. This is a serious mistake.
10. Please describe the microhardness marking correctly. Not HV, but HV0.1
Author Response
Comments and Suggestions for Authors
The article submitted for review concerns the microstructure and microhardness of the Inconel 718 alloy produced in additive technology. This alloy has been heat treatment.
- Abstract is written correctly, but I miss some details about the methodology. Please add 1-2 sentences about the conditions in which the samples were aged and what are the results of microhardness.
Response 1: Thanks for your suggestion. A new paragraph was included in the “Introduction” section.
- Specify the granularity of the powder, if you have it, attach a SEM photo of the powder.
Response 2: Dear reviewer, a full description of the powder has been incorporated in the “Materials and Methods” section. The powder granularity was included in a new figure, and the chemical composition was shown in a new table.
- What do classical metallographic procedures mean. Describe in detail the procedure for making a cross-section and the etching reagents.
Response 3: Actually, we follow the ASTM 7 standard. This information and others details of the metallographic procedures were included in the “Materials and Methods” section.
- Check the text carefully for typos, punctuation, and hardware data.
Response 4: Thanks for the observation. The entire document was reviewed.
- How the carbides were identified (were XRD studies carried out).
Response 5: Thanks for the comment. No, XRD studies were not performed. However, phase identification was based on the morphology, location, and composition of the precipitates observed by SEM and EDS. It is well known that carbides precipitate mainly at grain boundaries, and the morphology is blocky-like. Regarding the chemical composition, the carbides are rich in Nb, Ti, and C. In reference 17, we describe the complete methodology for phase identification by SEM and EDS. A new paragraph was included with this information in the “Materials and Methods” section.
- Please describe the microstructures in more detail, relate these descriptions to the manufacturing parameters.
Response 6: New paragraphs were introduced in the “3.1 Microstructure evolution” and the “Discussion” sections.
- Please relate your research results to other researchers in more detail.
Response 7: Dear reviewer, due to the lack of information available on reports related to microstructure evolution through heat treatment of AM-fabricated cellular-type structures, our results were compared with "bulk" samples obtained by AM. In this regard, the “Discussion” section was improved with a new text.
- There is no point in writing the size of the magnification (500x, 7500x etc.) It is enough that there is a scale bar in the photo. If someone prints a photo on half of a sheet of paper, the magnification data will not be correct. Therefore, remove this redundant data and leave only the scale bars.
Response 8: Dear reviewer, we appreciate your comment. The document was modified according to your observation. In addition, figures of the same sample (at 500X and 5000X) were merged to improve the clarity in the presentation of the results.
- How the linear EDS analysis was performed. The graphs look as if they were made from several points. You can't plot a graph from 5 points and represent it as linear EDS results. This is a serious mistake.
Response 9: Dear reviewer. We agree with you. The analysis of chemical composition was measured in several points along a line of 400 nm length. The measurements were taken every 100 nm, for this reason, there are only five data. However, they are enough to compare the concentration of the elements analyzed in these points. Consequently, the text was modified. Figures 13 and 14 of the original document were put together in a new figure, and the distance axes were aligned in all the figures.
- Please describe the microhardness marking correctly. Not HV, but HV0.1
Response 10: Thanks for the comment, it was a typing mistake. It was corrected the respective text as HV0.1
Reviewer 2 Report
In this work, the microstructure and 18 microhardness of cellular structures of Inconel 718, manufactured by additive manufacturing under 19 thermal treatments with different aging times, were evaluated.
Here you can find some major revision to improve article effort:
Introduction
The authors report that Despite the research done in bulk IN718, there is little information about cellular IN718 microstructure and the effect of heat treatment on this material. Please add citation and describe better what other authors have done and what results founded.
Materials and Methods
Please improve figure 1 with also cad drawn with a focus to the cell geometry and dimensions
Table 1 please add information about heating and cooling ramps that are still important
Results
3.2 Microhardness results
Table 2 and Figure 15 reports the microhardness results, Table 2 the standard deviation is not correct but in general all the values are constant, and this is impossible, moreover these values are different from what reported in figure 15.
Discussion
The authors report that Vickers microhardness values after heat treatment are much higher after 8 hours of 253 aging (540 HV) than other conditions (526 for 6 hours and 516 for 10 hours of aging). This must be proved with an AnoVA test coupled with Tukey range test applied to the results obtained from microhardness analysis
Author Response
Comments and Suggestions for Authors
In this work, the microstructure and microhardness of cellular structures of Inconel 718, manufactured by additive manufacturing under thermal treatments with different aging times, were evaluated.
Here you can find some major revision to improve article effort:
Introduction:
- The authors report that Despite the research done in bulk IN718, there is little information about cellular IN718 microstructure and the effect of heat treatment on this material. Please add citation and describe better what other authors have done and what results founded.
Response 1: Dear reviewer, new citations have been added in the “Introduction” section, and their results were discussed.
Materials and Methods:
- Please improve figure 1 with also cad drawn with a focus to the cell geometry and dimensions
Response 2: Thank you for your comment. Figure 1 was changed according to your suggestion, and a new paragraph was incorporated in the “Materials and Methods” section.
- Table 1 please add information about heating and cooling ramps that are still important
Response 3: Dear reviewer, the information on heating and cooling rates was added in one sentence before the corresponding table.
Results
Microhardness results
- Table 2 and Figure 15 reports the microhardness results, Table 2 the standard deviation is not correct but in general all the values are constant, and this is impossible, moreover these values are different from what reported in figure 15.
Response 4: Thank you for your observation. Unfortunately, it was a typo mistake from us. The values were corrected.
Discussion
- The authors report that Vickers microhardness values after heat treatment are much higher after 8 hours of 253 aging (540 HV) than other conditions (526 for 6 hours and 516 for 10 hours of aging). This must be proved with an AnoVA test coupled with Tukey range test applied to the results obtained from microhardness analysis
Response 5: This is an interesting observation. We included the ANOVA and Tukey range test to analyze the effect of each time for the first aging treatment. Two new tables were also included whit these calculations.
Reviewer 3 Report
I have a few comments which I have posted below:
- Line 52. There is: “δ-Ni3Nb” but 3 should be written in subscript.
- Line 56. There is: “IN718 [7, 13-¨15].” but should be: “IN718 [7, 13-15].”
- What was the chemical composition of the IN718 powder used?
- Line 123. There is: “a TUKON 1202 microhardness tester…” but should be: “a TUKON 1202 Vickers microhardness tester…”.
- The word "Laves" appears frequently in the text of the manuscript (e.g. lines 144, 146 etc.). However, it should be replaced with the phrase: "Laves phase/s".
- Why are the results of hardness measurements of the tested materials in Table 2 in the range of 318-368 HV, and in Figure 15 they reach the value of almost 540 HV?
Author Response
Comments and Suggestions for Authors
I have a few comments which I have posted below:
- Line 52. There is: “δ-Ni3Nb” but 3 should be written in subscript.
Response 1: Dear reviewer, we appreciate your observations. This suggestion was modified in the test.
- Line 56. There is: “IN718 [7, 13-¨15].” but should be: “IN718 [7, 13-15].”
Response 2: The document was modified according to your suggestion.
- What was the chemical composition of the IN718 powder used?
Response 3: complete information on the powder, including chemical composition, size distribution, and morphology, was included in the “Materials and Methods” section.
- Line 123. There is: “a TUKON 1202 microhardness tester…” but should be: “a TUKON 1202 Vickers microhardness tester…”.
Response 4: The document was modified according to your sugstions.
- The word "Laves" appears frequently in the text of the manuscript (e.g. lines 144, 146 etc.). However, it should be replaced with the phrase: "Laves phase/s".
Response 5: Dear reviewer, the document was modified according to your suggestions.
- Why are the results of hardness measurements of the tested materials in Table 2 in the range of 318-368 HV, and in Figure 15 they reach the value of almost 540 HV?
Response 6: Thank you for your observation. Unfortunately, it was a typo mistake from us. The values were corrected.
Round 2
Reviewer 1 Report
Thanks to the authors for the corrections made. I accept the article, but please consider the first review again.
Author Response
Point 1: Thanks to the authors for the corrections made. I accept the article, but please consider the first review again.
Response 1: Dear Reviewer. Thank you for your comment. The first review was considered again, and a few changes were made. The new document is uploaded in MS word template with the "track changes" function activated.
Reviewer 2 Report
All the correction were done. The paper can be accepted for publication
Author Response
Point 1: All the corrections were done. The paper can be accepted for publication
Response 1: Thank you for your comment.